# Chitosan Enhances the Anti-Biofilm Activity of Biodentine against an Interkingdom Biofilm Model

**DOI:** 10.3390/antibiotics10111317

**Published:** 2021-10-29

**Authors:** Sumaya Abusrewil, Jason L. Brown, Christopher Delaney, Mark C. Butcher, Mohammed Tiba, J. Alun Scott, Gordon Ramage, William McLean

**Affiliations:** 1Glasgow Endodontology Group, Glasgow Dental School, School of Medicine, Dentistry and Nursing, College of Medical, Veterinary and Life Sciences, Glasgow G12 8QF, UK; 2224354A@student.gla.ac.uk (S.A.); 2538282T@student.gla.ac.uk (M.T.); james.scott@glasgow.ac.uk (J.A.S.); gordon.ramage@glasgow.ac.uk (G.R.); 2Oral Sciences Research Group, Glasgow Dental School, School of Medicine, Dentistry and Nursing, College of Medical, Veterinary and Life Sciences, University of Glasgow, 378 Sauchiehall Street, Glasgow G2 3JZ, UK; Jason.brown@glasgow.ac.uk (J.L.B.); Christopher.delaney@glasgow.ac.uk (C.D.); 2135158B@student.gla.ac.uk (M.C.B.)

**Keywords:** endodontics, bioceramics, chitosan, biofilms, antimicrobials, interkingdom interactions

## Abstract

Endodontic infection is a biofilm disease that is difficult to irradicate with current treatment protocols, and as such, persistent micro-organisms may lead to ongoing or recurrent disease. The potential for the use of enhanced filling materials to modify biofilm regrowth is a promising strategy. This current study aimed to evaluate the anti-biofilm efficacy of calcium silicate cements modified with chitosan. The development of mono-species and multi-species biofilms on ProRoot MTA, Biodentine and bovine dentine discs were explored using quantitative microbiology analysis. The effect on regrowth of biofilms was assessed following the addition of chitosan to each cement. In comparison to a dentine substrate, both materials did not show the ability to inhibit biofilm regrowth. Biodentine incorporated with chitosan displayed a dose-dependent reduction in multi-species biofilm regrowth, unlike MTA. Notably, interkingdom biofilms were shown to enhance bacterial tolerance in the presence of chitosan. This study demonstrates the potential to enhance the antimicrobial properties of Biodentine. The findings highlight the need for appropriate model systems when exploring antimicrobial properties of materials in vitro so that interspecies and interkingdom interactions that modify tolerance are not overlooked while still supporting the development of innovative materials.

## 1. Introduction

Endodontic infection, in the form of biofilms, have been visualised colonising necrotic and treated root canals [1,2,3]. It has been evidenced that root canal infections exist as complex polymicrobial communities of bacteria and fungi [4]. Interkingdom interactions are highly relevant and should be considered in development of effective treatment strategies [4,5]. It is widely accepted that the chemo-mechanical means of disinfection during root canal treatment can be hampered by the intricate anatomy of the root canal system [6]. Therefore, despite chemo-mechanical disinfection, a significant challenge faced by dental cements used in the filling of the root canal space is the presence of persistent microorganisms [7].

In response to the challenges faced in sealing the root canal space, calcium-silicate-based materials have grown in prominence. The first member of the calcium-silicate-based materials to be introduced was mineral trioxide aggregate (MTA). MTA principally consists of tricalcium silicate, dicalcium silicate, tricalcium aluminate and tetracalcium aluminoferrite with bismuth oxide powder added as a radiopacifying agent [8]. Subsequently, a variety of new formulations of purer calcium-silicate-based materials have been developed based on tricalcium silicate chemistry [9]. These are termed bioceramics and are used primarily in endodontics [10]. Amongst this group of materials is Biodentine. These materials are indicated for a variety of endodontic procedures, including perforation repairs, regenerative endodontic procedures, retrograde obturation, vital pulp therapy and management of immature permanent teeth, similar applications to those outlined for MTA [11]. Previous studies have shown that these materials possess antibacterial and antifungal properties against isolated bacterial and fungal species. The antibacterial efficacy of the calcium silicate materials has been attributed to the alkaline environment formed when calcium silicate undergoes hydrolysis in water, producing calcium silicate hydrate and calcium hydroxide. The presence of precipitated calcium hydroxide results in an alkaline pH [12,13]. While this appears a useful function of the materials, there are limitations on the applicability of studies of this phenomenon. Many of these studies used material suspensions, in either media or sterile water, to test activity against only planktonic microbial cells, with the assessments based on determining minimum inhibitory concentrations (MIC) or suspension turbidity [14,15,16]. Such methods do not mimic the situation seen in clinical scenarios where it has been shown that microbes exist within biofilms, displaying unique phenotypic characteristics compared to their free-floating planktonic counterparts, including a notorious tolerance to antimicrobial agents under laboratory conditions [17,18]. Moreover, as highlighted, these studies use only mono-species systems that do not fully represent the polymicrobial nature of the infected root canal [19,20,21,22,23,24]. Confirming the importance of assessing the effect on biofilms, Jardine, Montagner [25], using an ex vivo biofilm model, demonstrated that calcium silicate materials were not effective against multispecies microcosm biofilms, even after 7 days of incubation. Therefore, our study suggests the need for augmenting materials with anti-biofilm active agents.

Chitosan is a modified natural carbohydrate polymer produced by deacetylation of chitin [26]. Chitosan is seen as a promising antimicrobial agent [27]. It is known to possess antimicrobial activities against a variety of Gram-positive and Gram-negative bacteria [28,29] and fungi [30,31]. The antimicrobial mechanism of action of chitosan is still unclear. However, it may be attributed to the affinity of the positively charged chitosan molecules for the negatively charged microbial plasma membrane which supports the interaction with anionic components of the cell membrane and leads to cell membrane disruption, intracellular contents leakage and ultimately cell death [31,32]. These properties lend themselves to the development of chitosan-based endodontic materials. Indeed, a previous study has highlighted the antibacterial effectiveness of chitosan nanoparticles (CNps) when incorporated into Ca(OH)_2_ pastes used as endodontic medicaments [33]. Other studies have shown that the antimicrobial effect of calcium-silicate-based sealers was enhanced by the incorporation of chitosan nanoparticles (CNps) in root canal sealers [29,34]. We and others have also shown the antimicrobial effectiveness of chitosan when used as a root canal irrigant [30,35,36]. Therefore, in the current study, we aimed to assess the anti-biofilm effects against complex polymicrobial biofilms of chitosan when incorporated into endodontic calcium silicate cements.

## 2. Results

### 2.1. Unmodified Calcium-Silicate-Based Materials Demonstrate Minimal Antimicrobial Effects in Comparison to Dentine

Biofilm regrowth and composition on dentine, MTA and Biodentine discs was assessed using live/dead qPCR. It was evident that both MTA and Biodentine did not show an ability to inhibit biofilm regrowth of any of the biofilm models after 24 h of incubation, compared to the control dentine discs (Figure 1A–C). Notably, the colony-forming equivalent (CFE) for *Candida* and bacteria formed on controls (dentine discs) were approximately 1 × log_10_ less when compared with colonies formed on the tested materials, although this did not reach statistical significance for viable cells.

When determining the potential effect of interkingdom interactions, there was approximately a 3.5-fold increase in viable bacteria when *C. albicans* was present on a dentine substrate (4.28 × 10^5^ compared to 1.48 × 10^6^ CFE), while on Biodentine, a much smaller 1.4-fold increase was apparent with respect to bacterial numbers in the absence of *C. albicans*. Meanwhile, no change was noted for bacterial loads on an MTA substrate (Figure 2). The results indicate that inclusion of *C. albicans* may support bacterial biofilm formation on a biological substrate. However, there were no or little supportive effects of *C. albicans* on bacterial numbers on abiotic surfaces. In contrast, on a dentine substrate, viable *C. albicans* showed a slight 1.2-fold increase when bacteria were incorporated, while an approximate 2-fold decrease was found in *C. albicans* CFE when grown on MTA and Biodentine in the presence of bacteria (Figure 2). These results might suggest some level of interkingdom antagonistic interactions with bacteria inhibiting *C. albicans* on abiotic surfaces. Total and live CFE counts for all biofilms and biofilm composition (%) for bacteria and *C. albicans* in each mixed biofilm are shown in Table 1.

### 2.2. Addition of Chitosan Confers Antimicrobial Properties on Biodentine, but Not MTA

The addition of chitosan to MTA imparted no antimicrobial enhancement against any of the biofilm models used (Figure 3A–C). Interestingly, in multispecies biofilms, CFE counts increased by 54.5% and 22% with the addition of 2.5% and 5% chitosan, respectively, when compared to unaltered MTA (Figure 3C). In contrast, for Biodentine, when *C. albicans* was grown as a mono-species biofilm, the live CFE following 2.5 wt% and 5 wt% chitosan incorporations was reduced by 83% and 71%, respectively, compared to the unmodified Biodentine. However, this reduction was not statistically significant (Figure 4A). In contrast, the addition of 2.5% and 5% chitosan reduced the live CFE/mL of the three-species biofilm model (bacteria only) by 85% and 97%, respectively, from 4.81 × 10^6^ CFE/mL (unmodified material) to 7.12 × 10^5^ and 1.42 × 10^5^ CFE/mL. The microbial reduction was dose-dependent, and the greatest reduction observed at 5% chitosan was significant (** *p* < 0.01). A decrease in bacterial load (*p* > 0.05) by 67% was also observed at 5% compared to the control (bovine dentine) (Figure 4B).

For the four-species biofilm model, adding 2.5% and 5% chitosan to Biodentine was able to effectively reduce live CFE by 56% and 90.5%, respectively, compared to the unaltered Biodentine (Figure 4C). The reduction from 8.24 × 10^6^ CFE/mL in the unaltered material to 3.6 × 10^6^ and 7.8 × 10^5^ CFE/mL in the chitosan-treated material (2.5% and 5%, respectively) was significant in the 5% added material (** *p* < 0.01). A decrease in the live CFE/mL of the mixed biofilm (*p* > 0.05) by ~55% was also observed at 5% compared to the control (bovine dentine). Of interest, the addition of chitosan preferentially targeted *C. albicans* in mixed-species biofilms and a concomitant significant (** *p* < 0.01) inhibition of regrowth of four-species biofilms (Appendix A). In contrast, at 5% chitosan, the reduction of bacteria number was not significant (*p* > 0.05) when *C. albicans* was present (Appendix A). However, in biofilms omitting *C. albicans*, a significant decrease in bacterial load was observed (** *p* < 0.01) at 5% chitosan (4B).

In a similar trend, the CFE count of *C. albicans* grown on Biodentine with 2.5% CNPs was decreased by approximately 2-fold when the three bacterial species were added. On the other hand, an approximate 12-fold reduction was noted for *C. albicans* on Biodentine with 5% CNPs (8.39 × 10^5^ compared to 6.79 × 10^4^), compared to *C. albicans* mono-species biofilm, in the presence of bacteria (Figure 5). However, the scenario was reversed with 4.7- and 5-fold increases of bacterial numbers observed at 2.5% and 5% CNPs, respectively, following inclusion of *C. albicans* (Figure 5). Live CFE counts for biofilms grown on Biodentine with 2.5% and 5% chitosan and biofilm composition (%) for bacteria and *C. albicans* in each mixed biofilm are shown in Table 2.

### 2.3. Addition of Chitosan Drives an Increase in pH for Biodentine but Not MTA

Both unmodified MTA and Biodentine cements exhibited an increase in alkalinity as setting proceeds. The pH values for MTA and Biodentine at 24 h were 12.7 and 11.5, respectively. However, elevated pH was determined for Biodentine when chitosan was incorporated and in a dose-dependent manner. The measurements of pH for MTA and Biodentine were approximately 12.8 and 12.6, respectively, at 24 h when 5% chitosan was incorporated into both cements (Figure 6). The pH of the manufacturer-supplied liquid component was also assessed for Biodentine and was determined to be 3.7.

## 3. Discussion

The introduction of calcium-silicate-based materials has been suggested as one of the most important advances in dental material science [37]. The antimicrobial properties of calcium-silicate-based materials have been widely investigated, though these studies are limited by their exploration of only single-species microbial models and the use of traditional microbiological techniques. The present study aims to close the gap in the literature, creating an understanding of the antimicrobial activities of these materials within a more relevant microbiological model system. *Streptococcus*, *Fusobacterium* and *Porphyromonas* are amongst the most frequently isolated bacterial species from endodontic infections [38]. *Candida* spp. has been shown to have a prevalence of 8.2% in endodontic infections [39]. Here, we were able to show for the first time that chitosan could be incorporated into Biodentine and effectively inhibit biofilm formation, opening the door for exploration of effective antimicrobial strategies for prevention and management of endodontic infection.

Our results indicated that neither material in an unmodified state shows an ability to inhibit biofilm regrowth of the three biofilm models after 24 h of incubation, compared to the control substrate of bovine dentine. The number of live *C. albicans* and bacterial colonies formed on both MTA and Biodentine were increased by approximately one log compared to those formed on dentine discs (Figure 1A,B), albeit without reaching statistical significance. This finding is interesting and raises the question as to whether dentine is demonstrating an antimicrobial effect in comparison to the cements, or if calcium silicate cements are supportive of microbial growth. Previous studies have demonstrated that dentine can enhance the antimicrobial effects of some materials and it has been hypothesised this is a result of changing the physicochemical nature of the materials with which dentine is combined [40,41]. However, in the present study, dentine was not combined with the materials. Therefore, it is not inconceivable that dentine itself as a biologically active substrate may have a weak antimicrobial capacity of its own. A previous study has demonstrated that the addition of a sterilised crushed human dentine to a suspension of *E. faecalis* did not exhibit any antibacterial activity [41]. However, in a separate study, it was shown that extracellular matrix isolated from the pulp and dentine of freshly extracted teeth demonstrated some level of antibacterial activity [42]. Another possibility is related to microbial adhesion. Type I collagen is the main organic component of dentine [43]. It has been suggested that collagen-rich substrates, such as dentine, can act as an ideal substrate for colonisation by *Streptococci* [44,45]. The preparation of dentine used in microbiological studies may have an effect on microbial adhesion [46]. In the present study, the high-temperature steam sterilisation used may have resulted in collagen denaturation, which has previously been suggested to reduce microbial adhesion [47]. In contrast, it has been demonstrated that dentinal collagen of bovine dentine slices, despite being partially denatured at high temperatures, can revert to its original confirmation [48]. Despite the potential confounding factors, it is felt that the use of dentine as a control substrate is appropriate. Previous studies have dispensed with biologically relevant substrates and used cell culture plastics as control surfaces for establishing the antimicrobial effect of this group of materials [19,20]. Although such in vitro biofilm systems have greatly enhanced our understanding of biofilm biology, their lack of biological and clinical relevance severely limits the understanding gleaned [49].

As stated, it was evident from the results that both unmodified ProRoot MTA and Biodentine were readily colonised by *C. albicans* and bacterial biofilms after 24 h of incubation. Neither material demonstrated antimicrobial properties against *C. albicans*, bacterial or interkingdom biofilms compared to the dentine control. It was clear from the findings that inclusion of bacteria inhibits *Candida* regrowth on both materials, whilst the addition of *C. albicans* showed a degree of enhancement of bacterial growth on these materials. Antagonistic interkingdom interactions have previously been highlighted where *F. nucleatum* and a number of other bacterial species including *Streptococci* and *P. gingivalis* inhibit growth and hyphal morphogenesis of *C. albicans* [50,51]. On the other hand, on the dentine substrate, the inclusion of *Candida* increased the number of viable bacteria by approximately 3.5-fold, while the addition of bacteria showed a small 1.2-fold increase in viable *C. albicans*. The mechanism may mirror the findings of Kean, Rajendran [52], albeit with *S. aureus,* where a strong synergy exists through the physical scaffold of hypha, providing a niche for colonisation a phenomenon that has been termed “mycofilms”. The microbial interactions differ according to substrate on which the interaction occurs. Such substrate-dependent phenomena have been described in other interkingdom interactions. Antagonistic interactions between *E. faecalis* and *C. albicans* in in vitro and in vivo models have been previously described [5,53]. However, Krishnamoorthy, Lemus [54] highlighted synergistic interactions between these species in an oral epithelium model.

The addition of chitosan to MTA provided no enhancement against the biofilm models after 24 h of incubation. In contrast, the combination of chitosan and Biodentine reduced the live colony-forming equivalent of the bacterial and mixed-species biofilms significantly. Notably, chitosan affected the composition of the evaluated four-species biofilms, causing a significant reduction in the viable fungal load in mixed culture. Interestingly, in the presence of *C. albicans*, bacterial load was decreased, but not significantly; in contrast, bacteria were decreased significantly when *C. albicans* was absent. These results indicate that fungi may confer protection to bacteria from active agents when grown in mixed microbial culture. This is in line with that described by Young, Alshanta [55], where protection to antimicrobial challenge is conferred upon bacterial species in the presence of *C. albicans*. It has also been shown that *C. albicans* ECM protected *S. aureus* against vancomycin treatment, possibly by limiting or delaying drug diffusion to *S. aureus* [56].

It is clear from our findings that ProRoot MTA and Biodentine exhibited different antimicrobial behaviours when chitosan was added. One mechanism by which calcium silicate cements have been said to exert an antimicrobial effect is through modifying environmental pH. It has been postulated that increased alkalinity, resulting from the release of calcium hydroxide upon setting of MTA preparations and its subsequent dissociation into calcium and hydroxide ions, may be responsible for any observed antimicrobial action [57,58]. To understand if the addition of chitosan modified pH, measurements of both materials’ leachate were taken. The unmodified MTA exhibited greater alkalinity than unmodified Biodentine at all time points assessed. This could be a result of differences observed in the pH of the manufacturer-supplied liquid components of both MTA (pH 7) and Biodentine (pH 3.7). It was also established that the addition of chitosan to MTA made no appreciable difference in pH at 24 h. However, upon addition of chitosan to Biodentine, a significant increase in pH was observed. This increase occurred in a dose-dependent manner. Given that the pH change merely brings it in line with that of MTA, it is unlikely that pH alone accounts for the antimicrobial activity differences seen between the two materials. However, in contrast to MTA, the increase in pH observed for modified Biodentine cement indicates that there may be an interaction between the cement components and the solid form of the chitosan particulate system. It has been shown that acidic chitosan solution displays a stronger antibacterial activity against *E. coli* than that of more alkaline solutions [59]. Other studies have shown higher antimicrobial activities when pH values of the chitosan solution ranged between 5 and 6.5–7; however, the inhibitory effect was completely abolished at pHs greater than 7 [60,61]. It has been suggested that the surrounding acidic medium leads to protonation of amino groups (NH_2_) of chitosan, which subsequently favours electrostatic interactions between the formed positively charged chitosan molecules and negative residues at biological sites [59,62]. Accordingly, the acidic pH of the manufacturer-supplied Biodentine liquid may have “activated” the chitosan, resulting in enhanced antimicrobial activity of the new compound.

## 4. Materials and Methods

### 4.1. Growing Multi-Species Biofilms

An established interkingdom endodontic biofilm model, previously described by our group, was used throughout this study [36]. Briefly, biofilms containing *Candida albicans* SC5314 (ATCC MYA-2876), *Streptococcus gordonii* (ATCC 35105), *Porphyromonas gingivalis* (ATCC 33277) and *Fusobacterium nucleatum* (ATCC 10953) were constructed. *C. albicans* was grown on Sabouraud’s dextrose agar (SAB) at 30 °C aerobically for 24–48 h; *S. gordonii* was grown on Columbia agar supplemented with 5% horse blood (CBA) at 37 °C in 5% CO_2_ for 24 h. The other two anaerobic organisms were maintained on fastidious anaerobic agar (FAA) plates containing 5% defibrinated horse blood at 37 °C in an anaerobic chamber (Don Whitley Scientific Limited, Bingley, UK) with an atmosphere of 85% N_2_, 10% CO_2_ and 5% H_2_ for 24–48 h. All agar bases were supplied by Oxoid, UK. Standardised cultures of *C. albicans* and bacteria (*S. gordonii*, *P. gingivalis* and *F. nucleatum*) standardised at 1 × 10^8^ CFU/mL were first diluted to 1 × 10^6^ CFU/mL and 1 × 10^7^ CFU/mL in culture broth, respectively. The broth consisted of 1:1 mixture of Roswell Park Memorial Institute-1640 (RPMI) with Todd Hewitt Broth (THB) supplemented with 0.01 mg/mL hemin and 2 µg/mL menadione. Four mixed-species biofilms were grown in pre-sterilised polystyrene 24-well flat-bottom plates (Costar^®^, Corning Incorporated, Corning, NY, USA) for 24 h in 5% CO_2_ at 37 °C. Two derived models were also used in parallel, one of which contained bacterial species only (*S. gordonii*, *P. gingivalis* and *F. nucleatum*) and one contained *C. albicans* only. This was to assess the importance of *C. albicans* in maintaining biofilm tolerance or otherwise.

### 4.2. Preparation of ProRoot MTA and Biodentine Materials ± Chitosan

Bovine dentine was used, which is an appropriate substitute for human dentine due to its availability and its great similarity to the human dentine [63]. Bovine dentine discs (Modus Laboratories, Reading, UK) were of 7 mm in diameter, 1 mm in thickness, with perpendicular dentinal tubule orientation (transverse cross section) and polished to 2500 micron on one side. The dentine discs were autoclaved at 122 °C, before use, for 16 min. Two bioceramic cements were used: mineral trioxide aggregate (MTA) (ProRoot MTA Root Repair Material (Dentsply Tulsa Dental Specialties, Johnson City, USA)) and Biodentine (Septodont, Saint-Maur-des-Fossés, France) (Table 3). Moulds, 1 mm in height with 7 mm diameter corresponding to the size of the bovine dentine discs, were fabricated from dental silicone-based impression materials; putty soft (Coltene, Altstätten, Switzerland); and polyvinyl siloxane impression material (Extrude, Romulus, MI, USA). The moulds were then disinfected with 70% ethanol. MTA and Biodentine powders were mixed according to the manufacturer. MTA powder was mixed with a ProRoot liquid micro-dose ampoule. The powder-containing capsule of Biodentine™ was mixed with 5 drops of the Biodentine™ Liquid.

To investigate the effect of chitosan on the antimicrobial properties of tested materials, chitosan was incorporated into MTA and Biodentine. Medium molecular weight Chitosan (CS (Sigma-Aldrich, St. Louis, MO, USA)) was used throughout this study. Briefly, chitosan powder was disinfected using UV for 15 min. Following this, the chitosan powder was incorporated into the Biodentine™ and MTA powders using two different concentrations (2.5 wt% and 5 wt%) and resultant powder was mixed with the manufacturer liquid component. Materials were then placed into aseptic moulds and allowed to set in a moist atmosphere at 37 °C for 3 h and 1 h, respectively.

### 4.3. Quantitative Analysis of Biofilms Formed on ProRoot MTA and Biodentine Materials ± Chitosan

The antimicrobial ability of ProRoot MTA and Biodentine was assessed against biofilm regrowth on the materials placed in a 24-well plate. Bovine dentine discs were used as positive controls. Four-species, three-species (bacteria only) and mono-species (*C. albicans* only) biofilms were grown in RPMI/THB in 24-well plates for 24 h, as previously described. After incubation, the spent biofilm media was discarded, and biofilms were washed with PBS, mechanically disrupted in 1 mL of media and diluted to 1:10 in fresh RPMI/THB and then inoculated on MTA and Biodentine discs (unaltered materials and altered ones with chitosan) into 24-well plates. Mechanical disruption of the biofilms serves the purpose of simulating mechanical debridement of the root canal. Plates were then incubated for an additional 24 h in 5% CO_2_ at 37 °C to allow biofilm growth.

Following incubation, each disc was washed with PBS, sonicated and transferred into a bijoux tube containing 1 mL PBS and then sonicated at 35 kHz in a sonic bath for 10 min. The sonicate was then transferred to 1.5 mL Eppendorf tubes (Greiner Bio-one, Kremsmünster, Austria, UK) for DNA extraction. The composition of the regrown biofilms on dentine, MTA and Biodentine discs was assessed using live/dead qPCR, a technique that uses propidium monoazide (PMA), a DNA-intercalating dye, to differentiate biofilm viable and dead microorganisms. Samples were prepared as previously described by [64]. Briefly, each sonicated sample was equally split; samples to be treated with PMA and control samples without PMA. Following this, 5 µL/mL of 50 µM PMA dye was added to each sample and incubated in the dark for 10 min. Treated and control samples were all incubated in the dark at room temperature for 10 min to allow cells to uptake the dye. Samples were then exposed to a 650 W halogen light and positioned 20 cm away from the sample tubes, for 5 min. During exposure, samples were placed on a bed of ice to avoid excessive heating. Following this, DNA extraction and real-time quantitative analysis were carried. Briefly, DNA was extracted from samples, according to manufacturer’s instructions, using the QIAamp DNeasy Mini Kit (Qiagen, Manchester, UK). Biofilm compositional analysis were enumerated using real time qPCR using SYBR^®^ GreenER™, with forward and reverse primers for either bacterial or *Candida* species, as listed in Table 4. qPCR was performed using the StepOnePlus™ Real-Time PCR system (Applied Biosystems, Waltham, MA, USA), and data analysed using the StepOnePlus software version 2.3 (ThermoFisher, Paisley, UK). All samples were run in duplicate with negative controls (master mix only) to assess for DNA contamination. The colony-forming equivalent (CFE) of samples was calculated using a previously established standard curve methodology [65] of serially extracted DNA bacterial and fungal colony-forming units from 1 × 10^4^ to 10^8^ CFU/mL. All experiments were performed three times, with three technical replicates.

### 4.4. Evaluation of pH of Leachate

Material discs 1 mm in thickness with 7 mm diameter were prepared of Biodentine and ProRoot MTA. Chitosan powder was incorporated into both materials at concentration of 2.5% and 5% by weight. The materials were allowed to set in a moist atmosphere at 37 °C for 3 h and 1 h, respectively. Each disc was then placed in a bijoux tube containing 3 mL sterile water. All samples were kept in a 5% CO_2_ incubator at 37 °C. Measurement of pH change of the storage solution was taken using a calibrated pH meter (Mettler Toledo, Leicester, UK) at the following time points: 1, 3 and 24 h. The pH measurement of the manufacturer Biodentine liquid was also taken.

### 4.5. Statistical Analysis

Data distribution, graphs and statistical analysis were performed using GraphPad Prism version 8 (GraphPad, San Diego, CA, USA). A D’Agostino-Pearson omnibus normality test was used before analysis to assess data distributions. Kruskal–Wallis with Dunn’s tests were used to determine the *p* values for non-parametric multiple comparisons (where data was not normally distributed). Differences were considered statistically significant when *p* < 0.05.

## 5. Conclusions

The present study demonstrates limited intrinsic antimicrobial abilities for the tested calcium silicate cements. This contrasts with previous studies but is likely a result of the use of multi-species biofilm models in the present study. This further highlights the need for the use of appropriate model systems in assessment of therapeutics to account for the often synergistic/protective relationships that exist in complex microbiological systems. The present study highlights the potential to enhance the biological properties of an existing calcium silicate cement, which may serve to reduce the likelihood of persistence or re-establishment of infections within the treated root canal space. Although some insight may have been gained into the mechanism through improvements in antimicrobial effects, further work is needed to fully elucidate this mechanism and understand the contribution of chitosan. Clearly, the modification of Biodentine to enhance antimicrobial properties has the potential to modify physical properties which may affect clinical handling and appropriateness. Further studies into material characteristics will be required to assess the use of chitosan as an active antimicrobial supplement in calcium silicate cements.

## Figures and Tables

**Figure 1 antibiotics-10-01317-f001:**
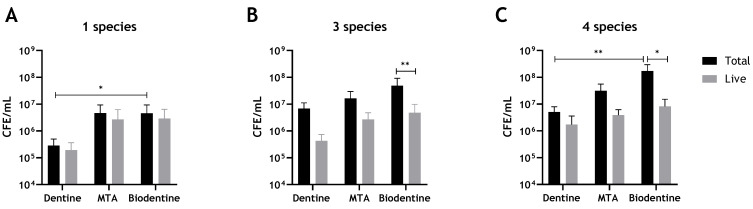
Compositional analysis of biofilms on MTA, Biodentine and bovine dentine discs. Biofilms were grown and assessed using live/dead qPCR. (**A**) Total and live CFE/mL of *C. albicans*-only biofilms, (**B**) Total and live CFE/mL of 3-species bacterial-only biofilms (*S. gordonii*, *P. gingivalis* and *F. nucleatum*). (**C**) Total and live CFE/mL of 4-mixed biofilms (bacteria and *C. albicans*). Data were analysed by Kruskal–Wallis with Dunn’s tests to determine the *p* values for non-parametric multiple comparisons. Differences were considered statistically significant when *p* < 0.05. * Indicates statistically significant differences (* *p* < 0.05, ** *p* < 0.01). Data representative of biofilms from three independent repeats n = 3 with 3 technical replicates.

**Figure 2 antibiotics-10-01317-f002:**
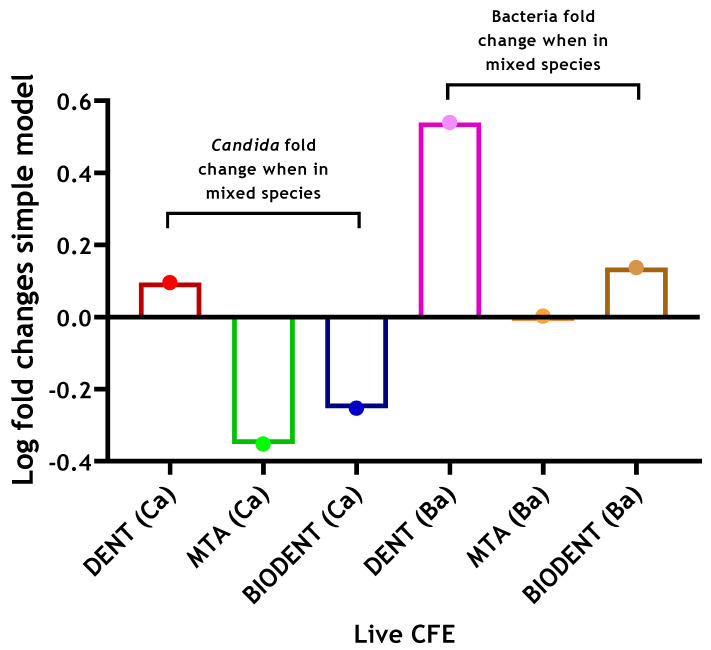
Log fold changes of live yeast and bacteria in complex 4-species biofilm model on three different materials. *C. albicans* and bacteria were quantified (CFE/mL) in mixed-species biofilms (4 species) and compared to simpler models of *C. albicans*-only biofilms (1 species) and bacterial-only biofilms (3 species), respectively. Log fold changes were calculated and presented graphically. Data representative from three repeats with 3 technical replicates.

**Figure 3 antibiotics-10-01317-f003:**
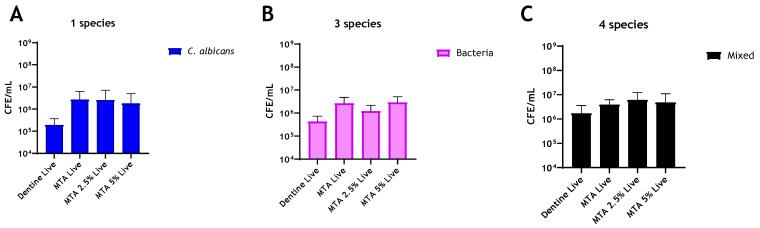
Compositional analysis of biofilms on MTA material discs. Live/dead qPCR was conducted following incorporation of 2.5 wt% and 5 wt% of chitosan: (**A**) Live CFE/mL of *C. albicans*-only biofilms, (**B**) Live CFE/mL of bacterial-only biofilms and (**C**) Live CFE/mL of 4-mixed biofilms (biofilms containing *C. albicans*). The bacterial and fungal loads were quantified using 16S and 18S primers, respectively. Bovine dentine and unaltered MTA discs were used as controls. Data were analysed by Kruskal–Wallis with Dunn’s tests.

**Figure 4 antibiotics-10-01317-f004:**
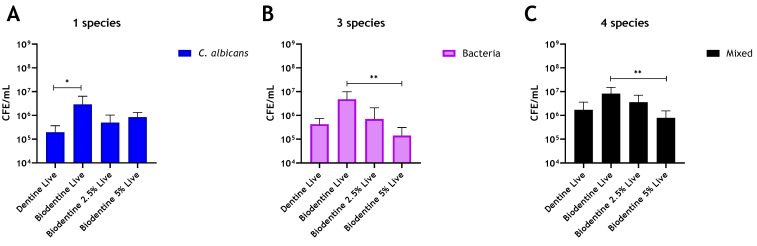
Compositional analysis of regrown biofilms on Biodentine material discs. Chitosan amounts of 2.5 wt% and 5 wt% were incorporated into Biodentine material, and live/dead qPCR was performed on (**A**) Live CFE/mL of C. albicans biofilms, (**B**) Live CFE/mL of bacterial biofilms and (**C**) Live CFE/mL of 4-mixed biofilms (bacteria and Candida). Bovine dentine and unaltered Biodentine discs were used as controls. Data were analysed by Kruskal–Wallis with Dunn’s tests. * Indicates statistically significant differences (* *p* < 0.05, ** *p* < 0.01).

**Figure 5 antibiotics-10-01317-f005:**
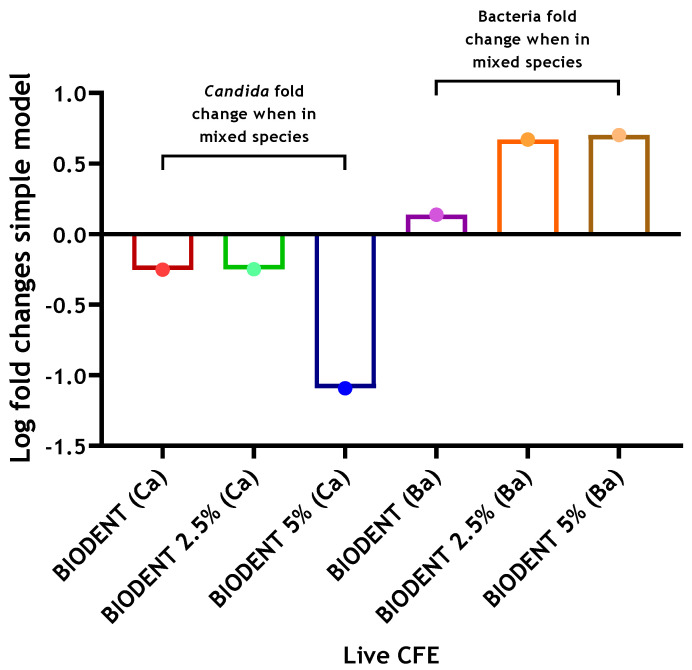
Log fold changes of live yeast and bacteria in complex 4-species biofilm model on Biodentine discs ± chitosan. *C. albicans* and bacteria were quantified (CFE/mL) in mixed-species biofilms (4 species) and compared to simpler models of *C. albicans*-only biofilms (1 species) and bacterial-only biofilms (3 species), respectively. Log fold changes were calculated and presented graphically. Data representative from three repeats with 3 technical replicates.

**Figure 6 antibiotics-10-01317-f006:**
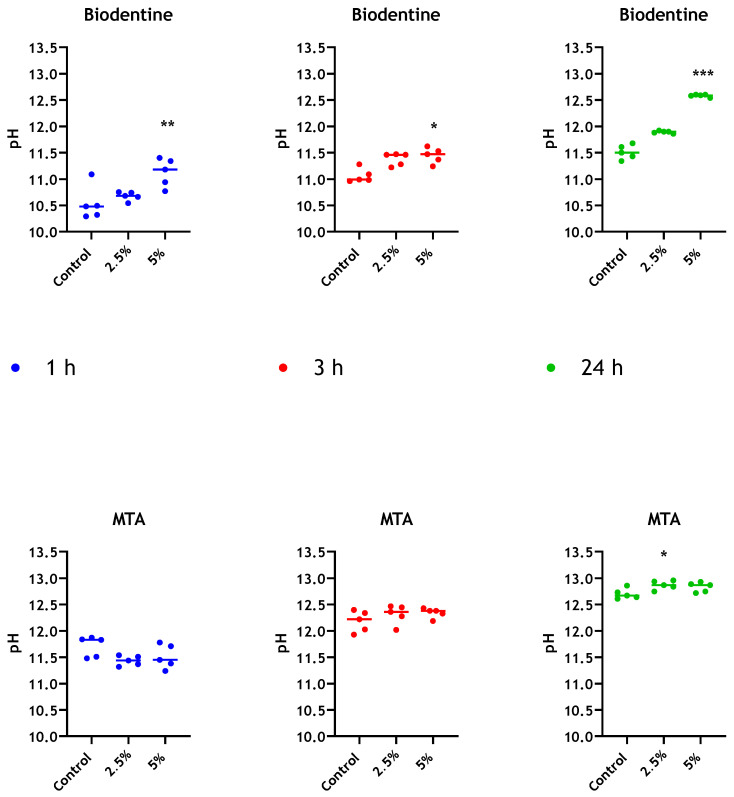
pH evaluation of leachate from different bioceramics. pH values for distilled water in contact with MTA and Biodentine + 2.5 wt% and 5% wt% CNPs, were determined at 1 h, 3 h and 24 h. Unmodified materials were used as controls. Data were analysed by Kruskal–Wallis with Dunn’s tests. Statistically significant differences between the control and each time point are presented as * *p* < 0.05, ** *p* < 0.01 and *** *p* < 0.001.

**Table 1 antibiotics-10-01317-t001:** Percentage composition of mixed-species biofilm model and total and live CFE counts for *C. albicans* and bacteria in all biofilms following biofilm growth on unaltered MTA, Biodentine and dentine substrates.

	4-Species Model				3-Species Model	Mono-Species Model
	Bacteria	(%) *	*C. albicans*	(%) *	Bacteria	*C. albicans*
Dentine (Total)	4.54 × 10^6^	90.21	4.92 × 10^5^	9.79	6.84 × 10^6^	2.85 × 10^5^
Dentine (Live)	1.48 × 10^6^	86.04	2.40 × 10^5^	13.96	4.28 × 10^5^	1.93 × 10^5^
MTA (Total)	2.90× 10^7^	92.0	2.53 × 10^6^	8.0	1.63 × 10^7^	4.62 × 10^6^
MTA (Live)	2.696 × 10^6^	69.17	1.20 × 10^6^	30.83	2.68 × 10^6^	2.70 × 10^6^
Biodentine (Total)	1.66 × 10^8^	97.01	5.10 × 10^6^	2.99	4.90 × 10^7^	4.52 × 10^6^
Biodentine (Live)	6.60 × 10^6^	80.16	1.63 × 10^6^	19.84	4.81 × 10^6^	2.92 × 10^6^

* Average percentage composition of bacteria and *C. albicans* in mixed-species biofilms.

**Table 2 antibiotics-10-01317-t002:** Live CFE counts for *C. albicans* and bacteria and percentage composition in all biofilms following biofilm growth on Biodentine discs.

	4-Species Model				3-Species Model	Mono-Species Model
	Bacteria	(%) *	*C. albicans*	(%) *	Bacteria	*C. albicans*
Biodentine (unaltered)	6.60 × 10^6^	80.16	1.63 × 10^6^	19.84	4.81 × 10^6^	2.92× 10^6^
Biodentine (2.5%)	3.33× 10^6^	92.22	2.81× 10^5^	7.78	7.124 × 10^5^	4.98 × 10^5^
Biodentine (5%)	7.120 × 10^5^	91.30	6.79 × 10^4^	8.70	1.42 × 10^5^	8.39 × 10^5^

* Average percentage composition of bacteria and *C. albicans.*

**Table 3 antibiotics-10-01317-t003:** Composition of ProRoot MTA and Biodentine.

Product	Composition	Manufacturer
White ProRoot Mineral Trioxide Aggregate (W-MTA)	Powder: tricalcium silicate, dicalcium silicate, bismuth oxide, tricalcium aluminate, calcium sulphate dihydrate or gypsum. Liquid: water	Dentsply Tulsa Dental Specialties, Johnson City, WA, USA
Biodentine	Powder: tricalcium silicate, dicalcium silicate, calcium carbonate, zirconium oxide, calcium oxide, iron oxide. Liquid: calcium chloride, a hydrosoluble (water-soluble) polymer, water.	Septodont, Saint-Maur-des-Fossés, France

**Table 4 antibiotics-10-01317-t004:** Primer sequences used for compositional analysis of biofilm models.

Organism	Primer	Forward Primer 5′-3′	Reverse Primer 5′-3′
*C. albicans*	18S	CTCGTAGTTGAACCTTGGGC	GGCCTGCTTTGAACACTCTA
Bacteria	16S	TCCTACGGGAGGCAGCAGT	GGACTACCAGGGTATCTAATCCTGTT

## Data Availability

Data is contained within the article or Appendix A.

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
