# Peer review of "Chitosan Enhances the Anti-Biofilm Activity of Biodentine against an Interkingdom Biofilm Model"

_antibiotics, 2021, doi:10.3390/antibiotics10111317_

Round 1
Reviewer 1 Report
The manuscript is well written and the topic of research is of significance in terms of controlling endodontic infections. However the following queries are to be addressed:
- Why chitosan addition to MTA showed no improvement in antimicrobial activity?
- How was the dosage of chitosan fixed at 2.5% and 5%?
- How do you justify the increase in CFE counts by the addition of chitosan in multispecies biofilms?
- Why chitosan addition increased pH in biodentine, but not in MTA?
Author Response
Many thanks for the helpful insight and critique of the submitted manuscript. Please find below response to the questions raised. Please note small changes have been made to correct some additional typographic errors.
- Why chitosan addition to MTA showed no improvement in antimicrobial activity? The finding that chitosan incorporation into MTA did not improve the antimicrobial activity may be due to the neutral pH of the manufacturer liquid component of MTA (pH 7). This is in contrast to the manufacturer liquid of Biodentine which was found to be acidic. Chitosan is soluble in dilute acidic medium and has shown greater antimicrobial activity in acidic environment in a number of studies. Thus "activation" of chitosan may have occurred leading to enhaced antimicrobial activity. We have highlighted this change on page 10, line 248-263.
- How was the dosage of chitosan fixed at 2.5% and 5%? Chitosan was studied at 2.5%, 5%, 10% and 20%. However, upon incorporation at the two higher percentages an obvious degradation of handling properties occurred. Thus, these percentages were excluded from further studies and results not included in the submitted manuscript.
- How do you justify the increase in CFE counts by the addition of chitosan in multispecies biofilms? The increase in CFE of bacterial species may result from fungi confering protection to bacteria from active agents when grown in mixed microbial culture. This effect is in line with other studies. We have highlighted this change on page 10, line 241-242.
- Why chitosan addition increased pH in biodentine, but not in MTA? The finding that chitosan incorporation into MTA did not change the pH of the material is may again be the result of the differences in the liquid componet of the material. Again, "activation" of chitosan may have occurred leading to altered interaction with Biodentine compared to MTA. We have highlighted this change on page 10, line 248-263.
Reviewer 2 Report
Reviewer comments
This manuscript describes “Chitosan enhances the anti-biofilm activity of biodentine against an interkingdom biofilm model”. This is interesting work highlighting the potential to enhance the biological properties of an existing calcium silicate cement. From this study, chitosan able to enhances the anti-biofilm activity of biodentine against bacterial and fungal biofilm models. This study can help the development of innovative materials.
This is useful work and can be consider for publication. Still, there are few shortcomings that will preclude its publication in the current form.
minor concerns:
- I would recommend authors to cite some more recent references in introduction and background section.
- Authors need to comment on the reason behind use of 2.5% and 5% w/w chitosan powder the Biodentine™ and MTA powders. Why these two conc is sufficient for this study to conclude chitosan utility as effective antimicrobial agents?
- Referencing is not uniform. Authors need to add all references in uniform pattern.
Manuscript can be considered for publication after these corrections.
Author Response
Many thanks for your helpful comments and review of the submitted manuscript. Please see the responses to your comments below. We have also identified and correct small typographical errors identified.
- I would recommend authors to cite some more recent references in introduction and background section. We have cited more recent references in introduction and background section (Ref number: 23-24-60-62).
- Authors need to comment on the reason behind use of 2.5% and 5% w/w chitosan powder the Biodentine™ and MTA powders. Why these two conc is sufficient for this study to conclude chitosan utility as effective antimicrobial agents? A greater range of concentrations were initially used in the study. These included 2.5%, 5%, 10% and 20%. However, it became apparent that at the hugher concentrations there was a significant degradation of material handling properties, so these were excluded from further analysis and not inclded in the submitted manuscript.
- Referencing is not uniform. Authors need to add all references in uniform pattern. We have added all references in a uniform pattern.